# MRI-Based Radiomic Machine-Learning Model May Accurately Distinguish between Subjects with Internet Gaming Disorder and Healthy Controls

**DOI:** 10.3390/brainsci12010044

**Published:** 2021-12-29

**Authors:** Xu Han, Lei Wei, Yawen Sun, Ying Hu, Yao Wang, Weina Ding, Zhe Wang, Wenqing Jiang, He Wang, Yan Zhou

**Affiliations:** 1Department of Radiology, Ren Ji Hospital, School of Medicine, Shanghai Jiao Tong University, Shanghai 200127, China; hanxu_ygritte@163.com (X.H.); cjs1119@hotmail.com (Y.S.); hu19700204@163.com (Y.H.); wangyao852004526@163.com (Y.W.); dingmeina1987@163.com (W.D.); 2Key Laboratory of Computational Neuroscience and Brain-Inspired Intelligence, Institute of Science and Technology for Brain-Inspired Intelligence, Fudan University, Shanghai 210023, China; weilei@fudan.edu.cn (L.W.); 16110840005@fudan.edu.cn (Z.W.); 3Human Phenome Institute, Fudan University, Shanghai 210023, China; 4Shanghai Mental Health Center, Department of Child & Adolescent Psychiatry, Shanghai Jiao Tong University, Shanghai 201109, China; jiangwenqing@sjtu.edu.cn

**Keywords:** magnetic resonance imaging, diffusion tensor imaging, radiomics, internet gaming disorder, random forest classifier

## Abstract

Purpose To identify cerebral radiomic features related to the diagnosis of Internet gaming disorder (IGD) and construct a radiomics-based machine-learning model for IGD diagnosis. Methods A total of 59 treatment-naïve subjects with IGD and 69 age- and sex-matched healthy controls (HCs) were recruited and underwent anatomic and diffusion-tensor magnetic resonance imaging (MRI). The features of the morphometric properties of gray matter and diffusion properties of white matter were extracted for each participant. After excluding the noise feature with single-factor analysis of variance, the remaining 179 features were included in an all-relevant feature selection procedure within cross-validation loops to identify features with significant discriminative power. Random forest classifiers were constructed and evaluated based on the identified features. Results No overall differences in the total brain volume (1,555,295.64 ± 152,316.31 mm^3^ vs. 154,491.19 ± 151,241.11 mm^3^), total gray (709,119.83 ± 59,534.46 mm^3^ vs. 751,018.21 ± 58,611.32 mm^3^) and white (465,054.49 ± 51,862.65 mm^3^ vs. 470,600.22 ± 47,006.67 mm^3^) matter volumes, and subcortical region volume (63,882.71 ± 5110.42 mm^3^ vs. 64,764.36 ± 4332.33 mm^3^) between the IGD and HC groups were observed. The mean classification accuracy was 73%. An altered cortical shape in the bilateral fusiform, left rostral middle frontal (rMFG), left cuneus, left parsopercularis (IFG), and regions around the right uncinate fasciculus (UF) and left internal capsule (IC) contributed significantly to group discrimination. Conclusions: Our study found the brain morphology alterations between IGD subjects and HCs through a radiomics-based machine-learning method, which may help revealing underlying IGD-related neurobiology mechanisms.

## 1. Introduction

While Internet use has made life easier, maladaptive Internet use can have unhealthy consequences, including psychological problems [1]. A widely accepted definition of Internet addiction is the excessive, uncontrolled, and harmful use of the Internet, and Internet gaming disorder (IGD) has been included in Section III of the Fifth Edition of Diagnostic and Statistical Manual of Mental Disorders (DSM-5) as a controversial term in 2013 (American Psychiatric Association (APA) 2013) [2]. Although there is a debate regarding whether Internet gaming could produce a true clinical addiction, emerging evidence shows that IGD subjects share similar neurobiological alternations with substance use and other behavioral addiction diseases such as gambling, particularly in craving, cognitive control, and reward systems. In 2019, IGD was included in the latest revision of the International Classification of Diseases (ICD-11; World Health Organization, 2019). Currently, the clinical diagnosis and evaluation of IGD are based on the integration of self-, parent-, and teacher-behavioral reports and the assessment of behavioral problems [3,4,5,6]. Given the subjective nature of these scales and the overlap of IGD with other psychiatric diseases, imaging-based parameters may provide a useful objective adjunct to the clinical evaluation of IGD [7,8]. Thus, it is crucial to establish a diagnosis model comprising biomarkers based on neuroimaging to accurately identify and characterize IGD. In the context of the developing field of psycho-radiology, machine learning is concerned with the automated discovery of regularities in brain imaging data through the use of pattern recognition algorithms to develop classifiers that can be used to predict disorders in individuals.

The radiomics approach is a medical image analysis framework that converts radiographic images into a mineable dataset using a series of data characterization algorithms. It has been applied to extract imaging features of solid tumors. Previous studies reported a 93% accuracy for a T1-weighted imaging (T1WI)-based support vector machine (SVM) algorithm for distinguishing between malignancy and benignity in soft-tissue lesions. Another study showed the high accuracy of a radiomics classifier based on fat-suppressed T2-weighted imaging. The radiomics workflow for oncology includes the extraction of quantitative imaging features from tumors to represent their intensity distribution, shape, and texture. Similarly, features can be extracted from different brain regions, such as the prefrontal cortex (PFC), anterior cingulate cortex (ACC), caudate, and internal and external capsules (IC and EC) [6,9,10,11,12]. Therefore, the radiomics workflow has also been applied to psychiatric disorders such as attention-deficit hyperactivity disorder (ADHD) [13,14].

In the current study, we applied a machine-learning method to construct an effective model to characterize IGD-related brain morphological changes.

## 2. Materials and Methods

### 2.1. Participants

The current study was approved by the Research Ethics Committee of Ren Ji Hospital and School of Medicine, Shanghai Jiao Tong University, China No. [2016]079k(2). All participants were informed of the aims of our study before MRI examination. Each participant submitted a written informed consent.

Between October 2016 and July 2017, 128 native Chinese-speaking right-handed young participants aged 13–28 years, including 59 participants with IGD and 69 HCs, were recruited for this study. All participants with IGD were recruited from the psychological outpatient clinic at the Shanghai Mental Health Center and were interviewed by two experienced psychiatrists. The criteria were assessed according to Young’s Diagnostic Questionnaire for Internet Addiction (YDQ) test, as modified by Beard [3]. The questionnaire consisted of eight “yes” or “no” questions, which were translated into Chinese. It included the following eight questions: (1) Do you feel absorbed in the Internet (remember previous online activity or the desired next online session)? (2) Do you feel satisfied with Internet use if you increase your amount of online time? (3) Have you failed to control, reduce, or quit Internet use repeatedly? (4) Do you feel nervous, temperamental, depressed, or sensitive when trying to reduce or quit Internet use repeatedly? (5) Do you stay online longer than originally intended? (6) Have you taken the risk of losing a significant job, relationship, educational, or career opportunity because of the Internet? (7) Have you lied to your family members or others to hide the truth of your involvement with the Internet? (8) Do you use the Internet as a way to escape from problems or relieve an anxious mood? Respondents who answered “yes” to questions 1 through 5 and at least one of the remaining three questions were classified as having Internet addiction. For a more controlled and homogenous IGD sample, only those participants who reported playing massive multiplayer online role-playing games, such as World of Warcraft, as their main use of the Internet were selected. We confirmed the reliability of the self-reports by talking with participants’ parents. In addition, we evaluated the severity of IGD using the Chen Internet Addiction Scale (CIAS), which is a self-reported scale with good reliability and validity that has been used to measure the severity of Internet addiction [15]. The questionnaire contains 26 items answered on a four-point scale for which a diagnostic cutoff point (63/64) exhibited the best diagnostic accuracy. The HC participants were recruited through advertisements in the community. These participants were also tested using the modified YDQ criteria and none met the diagnostic criteria for IGD. We also used the Behavioral Impulsive scale-11 (BIS-11) to assess behavioral inhibition. The BIS-11 is a questionnaire consisting of 30 items designed to assess the personality/behavioral construct of impulsiveness [16]. The self-rating anxiety scale (SAS) and self-rating depression scale (SDS) were used to show that all the subjects met the inclusion criteria during the research period. All the questionnaires were initially written in English and then translated to Chinese.

None of the participants had (1) previous hospitalization for psychiatric disorders or a history of psychiatric disorders such as anxiety, depression, or attention-deficit hyperactivity disorder; (2) substance use disorders; (3) mental retardation; (4) neurological illness or injury; and (5) intolerance to magnetic resonance imaging (MRI). All participants were evaluated using brain MRI.

### 2.2. MRI Acquisition

Images were obtained using a 3.0 T MRI scanner (Signa HDxt 3T, GE Healthcare, Milwaukee, WI, USA). Restraining foam pads were used to reduce head motion, and earplugs were used to reduce scanner noise. 3D Fast spoiled Gradient Recalled sequence (3D-FSPGR) images (TR = 5.6 ms, TE = 1.8 ms, slice thickness = 1 mm, gap = 0, flip angle = 15°, FOV = 256 mm × 256 mm, number of slices = 156, 1 × 1 × 1 mm voxel wise) and DTI data (TR = 17,000 ms, TE = 89.8 ms, slice thickness = 2 mm, gap = 0, FOV = 256 mm × 256 mm, number of slices = 66, matrix = 128 × 128, and 20 diffusion-weighted directions with b value = 1000 s/mm^2^) were acquired; a reference image with no diffusion gradients applied (B0 scan) was also acquired.

The following sequences were also performed to confirm the absence of structural lesions: (1) axial T1-weighted fast field echo sequences (TR = 331 ms, TE = 4.6 ms, FOV = 256 × 256 mm^2^, matrix = 512 × 512, thickness = 4 mm, gap = 0, slices = 34); (2) axial T2-weighted turbo spin-echo sequences (TR = 3013 ms, TE = 80 ms, FOV = 256 × 256 mm^2^, matrix = 512 × 512, thickness = 4 mm, gap = 0, slices = 34). All images were evaluated by two experienced neuroradiologists and no participants were excluded on this basis.

### 2.3. MRI Data Pre-Processing and Feature Extraction

The flowchart for this study is described in Figure 1.

The T1-weighted anatomical images were first processed by Freesurfer software (Version 6.0) with the “recon-all” processing pipeline, which included skull stripping, image registration, subcortical segmentation, cortical surface reconstruction, cortical segmentation, and cortical thickness estimation. All processed images were visually inspected to avoid skull strip failure, segmentation errors, and topology failure. The images were labeled using the Desikan–Killiany–Tourville atlas [17]. For extraction of gray matter features, the labeled T1-weighted images were first converted to a surface mesh; the shape properties were then calculated on each vertex and the distribution metrics were generated (mean, standard deviation, kurtosis, skewness). A total of 1316 features related to gray matter morphometry were calculated using Mindboggle. The feature types included volume, surface cortical thickness, geodesic depth, convexity, and travel depth.

Diffusion tensor images were implemented using a pipeline toolbox, PANDA v1.3.1 (https://www.nitrc.org/projects/panda, accessed on 23 May 2020), which is based on FSL tools [18]. In the pipeline, skull stripping with the brain extraction tool (BET) was performed to extract brain tissue from the B0 images in each subject. Eddy current-induced distortion and head motion artifacts were corrected by registering each raw diffusion image to the B0 image with an affine transformation. Four calculated parameters, namely fractional anisotropy (FA), mean diffusivity (MD), radial diffusivity (RD), and axial diffusivity (AD), were created. The JHU-ICBM-81 atlas was warped to each individual space and the distribution metrics were generated (mean, standard deviation, kurtosis, skewness) for each label. Finally, 768 features representing the diffusion properties were extracted.

### 2.4. Feature Selection and Assessment of the Relevance of the Selected Features

The set of features represented both the gray and white matter profiles of each brain (1316 Gy matter and 768 diffusion features, for a total of 2084 features). All of the extracted features were normalized by z-score. The “Boruta” (https://www.rproject.org/, accessed on 17 July 2019) algorithm was used to extend the given dataset by appending shuffled copies of all features, which are called shadow features. Then, a random forest (RF) classifier was trained in the extended dataset and the relevance of real features was evaluated by comparing the importance measure provided by RF between the real and shadow features. Before feature selection, one-way analysis of variance was used to exclude the noise feature. A total of 179 features eventually remained after the coarse filter. At each iteration, the algorithm checked whether a real feature had a higher relevance to classification than the best of its shadow features and removed features that were deemed irrelevant to classification. The algorithm terminated when the relevance of all the features was established. The features were grouped into two categories: relevant and irrelevant.

In our study, features were selected for the different subsets in each cross-validation iteration. Features that were selected in more iterations than would be expected to occur at random were identified as significantly relevant selections. To determine the relevance of the selected features, 1000 datasets with a random permuting label column were created. The expected distribution of the selection frequency (defined as the number of iterations in which a feature was selected divided by the total number of iterations performed in one data set) of each feature throughout the cross-validation iterations was modeled as a binomial distribution, with the parameter estimated as the mean selection frequency in all random data sets. This distribution was then used to identify features in the original dataset with selection frequencies significantly higher than expected by chance, with adjusted *p* values of 0.05 (Bonferroni correction). To control for the influence of low selection frequency in the original datasets, we further applied a threshold on the original feature selection frequency. Selected features lower than half of the total iterations in the original dataset were excluded.

### 2.5. Construction of the RF Classifier

RF was used to build classifiers in our study because it has demonstrated superior performance on high-dimensional, low sample size problems and requires little feature processing and parameter tuning [19]. The all-relevant feature selection step was embedded in a repeated k-fold (*k* = 5) cross-validation framework to obtain unbiased estimates of the classification error. The R package “caret” (classification and regression training) was used to implement this procedure [20]. In each cross-validation loop, the entire dataset was randomly partitioned into five non-overlapping five-folds of equal size. Four folds were entered into the all-relevant feature selection procedure and the reduced dataset with selected features was used to train an RF classifier. We considered the default parameter configuration for the value of ntrees (number of trees) equal to 1000 and the mtry (number of features randomly selected at each tree node) equal to the root of the number of input features. The remaining fold, simplified with selected features, was used to evaluate the performance of the model. The selection–training–testing cycle was repeated for different left-out portions. This entire loop was repeated 100 times with different partitioning schemes to achieve stable performance estimation. The overall accuracy, sensitivity, specificity, and k score were used to characterize the performance of the classifier.

### 2.6. Statistical Analysis

All data analyses and statistics were performed using R-3.6.0 (https://www.r-project.org, accessed on 8 October 2020). Kolmogorov–Smirnov tests were used to test the distributions of age, education, and identified features. Normally distributed data were compared using *t*-tests, while nonparametric tests were used for non-normally distributed data. Chi-square tests were used to compare sex between the two groups. Statistical significance was set at a two-tailed *p*-value of 0.05.

## 3. Results

### 3.1. Demographic and Volumetric Comparison

The demographic variables and macroscopic cerebral volume are shown in Table 1. No significant differences in sex, age, years of education, total gray matter volume, total white matter volume, total subcortical volume, and total brain volume were found. The IGD group scored higher on BIS-11, CIAS, SAS, and SDS (all *p* < 0.0001) (Table 2).

### 3.2. Classification Performance and Significantly Relevant Features

After Boruta selection in the training set, eight features were identified as significantly relevant for a selection frequency in the real data that was significantly higher than that in the random data (Table 3, Figure 2 and Figure 3). The classification accuracy and k values were 0.73 ± 0.08 and 0.45 ± 0.16, respectively, with features from the all-relevant feature selection step. The sensitivity and specificity were 0.77 ± 0.11 and 0.68 ± 0.13, respectively.

## 4. Discussion

In this study, we employed RF to discriminate between subjects with IGD and HCs with a 73% accuracy. More importantly, alterations in rMFG, fusiform, cuneus, IFG, IC, and UC were identified, contributing to the models during the all-relevant feature selection process. Previous studies have constructed machine learning models to discriminate between IGDs and HCs using neuroimaging data. Song et al. employed a modified connectome-based predictive model (CPM) to identify resting-state connections associated with IGD [1]. Although they achieved higher accuracy (78.76%), the lack of diffusion-weighted MR imaging data may have limited the efficiency of the constructed classifier because many previous studies have reported white matter abnormalities. Our study is the first attempt to characterize brain morphological changes of IGD subjects using a radiomics-based classification model considering the properties of both gray and white matter, providing a more comprehensive understanding of the neuroanatomical alterations related to IGD. The same analysis approach has been successfully applied for the diagnosis of attention-deficit hyperactivity (ADHD) [14].

Our findings demonstrate that gray matter abnormalities in individuals with IGD were most frequently located in the MFG, IFG, cuneus, and fusiform regions, which are reportedly related to cognitive control, decision making, and reward processing [6,11,21]. Curvature, local thickness, and depth are measures frequently used to characterize cortical folding patterns. Bilateral fusiform was important in the present classifier. Previous neuroimaging studies [22,23,24] have reported similar findings and indicated that the activated fusiform gyrus is associated with game craving, semantic processing, disembodiment, and working memory [22]. Local alterations in thickness may further reflect inner impaired function. In the human brain, the morphologies of the cortical gyri and sulci are complex and variable among individuals, which may cause and reflect abnormal function. Local cortical thickness, convexity, curvature, and depth are frequently used shape analysis features for characterizing cortical folding patterns. In the present study, we also found that IGD was strongly associated with morphometric features in regions of the MFG and IFG, which play vital roles in frontal-striatal circuits [11,25,26]. In particular, the frontal lobe is involved in inhibitory control [27,28,29], which can be easily affected by the long-term use of Internet games [30]. Abnormal resting-state functional connectivity within frontal-striatum circuits was observed in IGD in our previous studies, which supports the results of the present study [26]. Moreover, this study proposed an automated, convenient workflow to provide a potentially useful method for revealing the brain morphological alternations in individuals with IGD. The cuneus is considered a visual processing and inhibitory control center [31]. Previous neuroimaging studies in substance-dependent individuals have observed abnormalities in the cuneus [32]. Pezawas et al. also reported decreased regional cerebral blood flow in the occipital cortex, including the bilateral cuneus, in heroin-addicted subjects [33]. Thus, we postulate that structural abnormalities in the cuneus may be partly responsible for the deficits in the inhibitory control in IGD.

The IC runs from the thalamus to the frontal cortex [34] and is reportedly involved in reward circuits. However, this association remains speculative and warrants further investigation [35]. Previous studies have reported abnormalities in IC integrity in individuals with addiction, while our study results indicated that the standard deviation of the MD changed in the IC [36]. MD is the mean extent of the three-directional diffusion, which measures the average diffusion of water molecules within tissues, whereas the trace is the sum of the three directional diffusions. Thus, an increased MD in white matter generally indicates disruption of white matter microstructures [37]. Moreover, the standard deviation can reflect the degree of dispersion of a dataset, and the dispersion of the MD may further influence the function across the entire IC fiber tract. We also found abnormal MD in the UF, a white matter tract critical for frontal-temporal lobe functional integration; moreover, the results of the current study suggested that this circuit involving the frontal, temporal lobe, and UF may be linked to the cerebral dysregulation associated with IGD [38]. Assuming that greater white matter consistency promotes the transmission of functional information, higher consistency might regulate brain activity in projection target regions and even their functional couplings associated with IGD.

Several limitations of this study should be considered when interpreting the data. First, our sample size was relatively small, which might have limited the generalizability of our results. Additional studies with larger, independent, and multicenter datasets are needed to confirm our findings. Second, our study used the JHU and DKT atlas. However, previous studies have demonstrated that different parcellation schemes or spatial scales generate different results. Further studies should determine which brain atlas is appropriate for classifying IGD and HCs. Besides, DTI measurement is limited by the linearity of the diffusion sensitizing gradient. The deviation between the real and measured orientation of fibers is directionally dependent, what was confirmed in MRI measurement. The deviation errors can be effectively corrected by preceding the DTI measurement with the b-matrix Spatial Distribution in DTI (BSD-DTI) calibration [39]; however, we have missed consideration of the potential systematic errors. In the future, we will try the BSD-DTI. Moreover, we only recruited subjects aged 13–28 years; although this population is particularly at risk for IGD, the current results are specific to younger individuals with IGDs, and additional studies are needed to verify our findings in other age groups.

## 5. Conclusions

Our study found the brain morphology alterations in the regions of rMFG, fusiform, cuneus, IFG, IC and UC between IGD subjects and HCs through a radiomics-based machine-learning method, which may help revealing underlying IGD-related neurobiology mechanisms.

## Figures and Tables

**Figure 1 brainsci-12-00044-f001:**
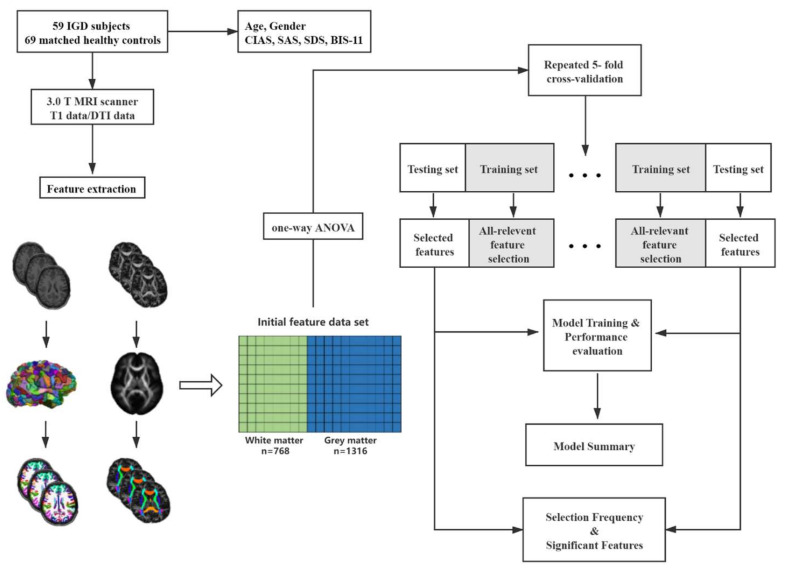
Flowchart of this study. IGD = Internet gaming disorder; CIAS = Chen Internet Addiction Scale; BIS-11 = Behavior impulsive scale-11; SAS = Self-rating anxiety scale; SDS = Self-rating depression scale.

**Figure 2 brainsci-12-00044-f002:**
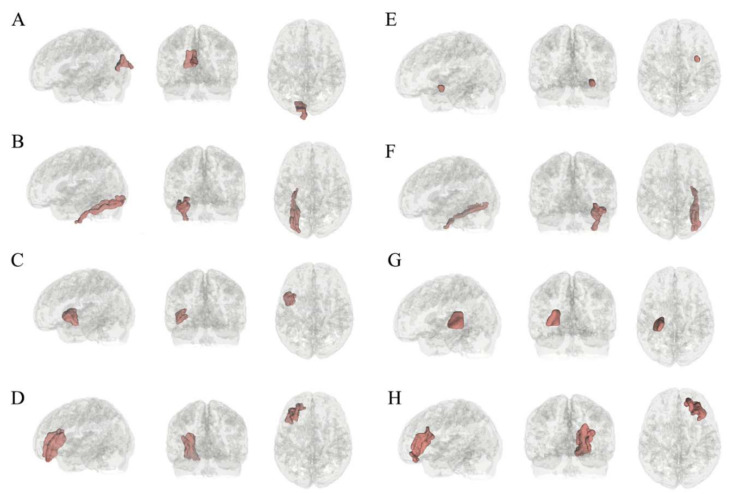
Identified features that discriminated IGD subjects and HCs. IGD = internet gaming disorder; HC = healthy control; (**A**) left cuneus; (**B**) left fusiform; (**C**) left parsopercularis; (**D**) right rostral middle frontal; (**E**) right uncinate fasciculus; (**F**) right fusiform; (**G**) left internal capsule; (**H**) left rostral middle frontal.

**Figure 3 brainsci-12-00044-f003:**
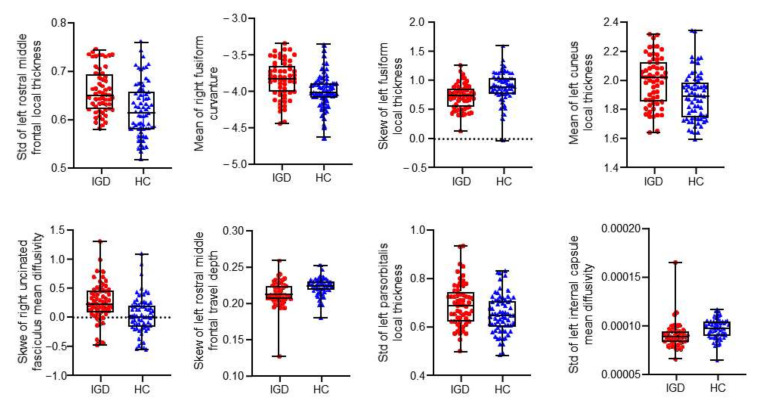
Identified features that discriminated IGD subjects and HCs. IGD = internet gaming disorder; HC = healthy control; std = standard deviation.

**Table 1 brainsci-12-00044-t001:** Demographic statistics.

	IGD	HC	Statistic	Degree of Freedom	*p* Value
**Age**	21.39 ± 3.06 (15–28)	20.34 ± 3.98 (13–28)	1.639	126	0.104
**Gender**			0.417 χ^2^	1	0.518
**Male**	47	58			
**Female**	12	11			
**Total gray matter volume (mm^3^)**	709,119.83 ± 59,534.46	751,018.21 ± 58,611.32	−0.563	126	0.574
**Total white matter volume (mm^3^)**	465,054.49 ± 51,862.65	470,600.22 ± 47,006.67	0.634	126	0.527
**Subcortical region volume (mm^3^)**	63,882.71 ± 5110.42	64,764.36 ± 4332.33	−1.056	126	0.293
**Total brain volume (mm^3^)**	1,555,295.64 ± 152,316.31	15,4491.19 ± 151,241.11	0.03	126	0.976

IGD: Internet gaming disorder; HC: Healthy control.

**Table 2 brainsci-12-00044-t002:** Scales.

	IGD	HC	Statistic	Degree of Freedom	*p* Value
**CIAS**	78.27 ± 10.31	44.38 ± 11.34	17.57	126	<0.0001 *
**BIS-11**	63.02 ± 7.72	53.81 ± 7.42	6.87	126	<0.0001 *
**SAS**	50.51 ± 8.19	42.65 ± 6.39	6.09	126	<0.0001 *
**SDS**	51.97 ± 7.09	45.74 ± 8.92	4.32	126	<0.0001 *

IGD: Internet gaming disorder; HC: Healthy control; CIAS: Chen Internet Addiction Scale; BIS-11: Behavior impulsive scale-11; SAS: Self-rating anxiety scale; SDS: Self-rating depression scale; * *p* < 0.05.

**Table 3 brainsci-12-00044-t003:** Significant features for discriminating IGD and HC.

Selection Frequency (%)	Hemisphere	Label	Feature Type	Statistic	IGD *	HC *
99.6	Left	Rostral middle frontal	Local thickness	Standard deviation	0.66 ± 0.05	0.62 ± 0.06
96.8	Left	Internal capsule	Mean diffusivity	Standard deviation	0.000089 ± 0.000013	0.000095 ± 0.000010
84.0	Right	Fusiform	Mean curvature	Mean	−3.83 ± 0.26	−4.00 ± 0.23
83.8	Left	Fusiform	Local thickness	Skewness	0.70 ± 0.22	0.86 ± 0.26
83.2	Left	Cuneus	Local thickness	Mean	1.99 ± 0.17	1.88 ± 0.16
77.8	Right	Uncinate fasciculus	Mean diffusivity	Skewness	0.25 ± 0.33	0.03 ± 0.32
74.4	Left	Rostral middle frontal	Travel depth	Skewness	0.21 ± 0.02	0.22 ± 0.01
72.6	Left	Parsorbitalis	Local thickness	Standard deviation	0.52 ± 0.05	0.49 ± 0.06

* Data are means ± standard deviation; IGD: Internet gaming disorder; HC: Healthy control.

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
