# Peer review of "MRI-Based Radiomic Machine-Learning Model May Accurately Distinguish between Subjects with Internet Gaming Disorder and Healthy Controls"

_brainsci, 2021, doi:10.3390/brainsci12010044_

Round 1
Reviewer 1 Report
if file

Reviewer 2 Report
Overview:
The authors present an interesting radiomics scheme for discovering potential neuroimaging features associated with Internet gaming disorder (IGD). The paper is well written and easy to follow. However, I have the following concerns that require extra iterations over the work.
Comments:
- What are the features that belong to the white matter regions? In the paper, the authors described that their radiomic features include morphological ones derived from the white and grey matter regions. But in section 2 I understood it is 1316 Gy matter and 768 diffusion features.
- Feature normalization/standardization is not used in the paper, this should be explained and justified. In any feature selection pipeline, this is a crucial step to ensure feature relevance is not corrupted by improper scaling.
- It is interesting to see region-based analysis (independent random-forest classifiers built on each brain region exploiting structural (T1) and physiological (DTI) features. I think this way is interesting to converge into a more solid conclusion that independently correlates changes of morphological/diffusion parameters across different regions of the brain with IGD.
- Brain neuroimaging features used in the analysis are pooled regardless of their spatial location (no correspondence was tracked with brain regions). I believe that there should be more details explaining how the set of features after selection were linked with brain regions in Fig 2. I expect that if a brain region is relevant to IDG, then all features in that region should relevant too, is that correct?
- Section 2.6 seems to be a repetition or an extra explanation of the Boruta algorithm mentioned in Section 2.4. If this is the case, I think that these two sections should be merged. Otherwise, better structure and explanation distinguishing the analysis done in both sections should be there.
- I think is important to show the statistical difference that can be done as a box plot overlaid with dot/scatter plot showing feature values of control and study groups in Fig 3.
- Is Figure 2 resulting from analyzing features from both structural and diffusion MRI? Again, my point here is that this region-based relevance to IGD should be better explained in the paper.
Round 2
Reviewer 1 Report
1. TE time can't be more than TR, please correct it effectively.
2. Briefly explain the meaning of the BSD-DTI and add the appropriate references.
3. The sentence structures are sometimes incorrect, please correct the text using the native speaker.
The manuscript is better, revised as above, possible to publish.
Good luck with your further research
Author Response
please see attached, thank you
